# Resource Allocation for TDD Cell-Free Massive MIMO Systems

**Xuanhong Lin, Fangmin Xu \*, Jingzhao Fu and Yue Wang**

Institute of Communications Engineering, Hangzhou Dianzi University, Hangzhou 310018, China; 18081517@hdu.edu.cn (X.L.); 192080089@hdu.edu.cn (J.F.); yuewanghdu@yeah.net (Y.W.)
\* Correspondence: xufangmin@hdu.edu.cn

**Abstract:** In this paper, we investigate a joint resource allocation algorithm in a time-division duplex (TDD)-based cell-free massive MIMO (CFMM) system, which has great potential to improve spectrum efficiency and throughput. Because the throughput of the system is a bottleneck due to the sharing of the pilot, we attempted to alleviate pilot contamination. We propose a pilot assignment approach called user-distance-ordering-based pilot assignment (UDOPA) based on the distance between users and the center, which can be calculated by the K-means method. Then, using an access point (AP) selection algorithm, only the APs having a major impact on the macro diversity gain of a user are selected as the serving APs. In contrast to the existing AP selection algorithms, users with the same pilot are not allowed to share the same serving AP in the proposed AP selection algorithm, which also significantly reduces the complexity of data processing. Finally, a modified max–min power control scheme with teaching–learning-based optimization (TLBO) is proposed to further improve the performance of the systems and guarantee the minimum user rate. Simulation results show that the proposed joint resource allocation scheme can effectively enhance CFMM systems' performance.

**Keywords:** cell-free massive MIMO (CFMM); pilot contamination; AP selection; power allocation

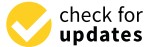



## 1. Introduction

### 1.1. Motivation

Recently, cell-free massive MIMO (CFMM) technology has attracted a large amount attention because it has great potential to improve cell-edge user performance, spectrum efficiency, and throughput. In this system, a large number of distributed access points (APs) are spread across a wide field and serve many users using the same time-frequency resources. In contrast to distributed antenna systems, the number of users is much smaller than that of APs in the CFMM [1–7]. Compared with the cellular system, users will be served more evenly in the CFMM system. However, the pilot contamination greatly limits the performance of time-division duplex (TDD)-based CFMM systems. Pilot allocation effectively alleviates pilot contamination and has been extensively studied for massive MIMO systems [8,9]. Notably, CFMM is different from massive MIMO [10] in terms of many aspects, such as channel hardening [11] and spatial correlation [12]. Therefore, pilot allocation based on traditional information theory for massive MIMO systems [13–15] is not applicable to CFMM systems. To tackle this problem, a large number of studies have proposed resource allocation schemes, such as pilot assignment and power control, for TDD CFMM systems. However, the performance of these schemes still needs to be improved.

To alleviate the performance degradation caused by the pilot contamination, we investigated resource allocation problems in CFMM systems, including pilot allocation, AP selection, and power control. Firstly, to mitigate the problem of pilot contamination, the pilot assignment is based on the order of user distance, in which pilot sharing is only allowed when the distance between two users is large enough to reduce the impact of pilot reuse. Then, using AP selection, only the APs having a major impact on the macro diversity gain are selected as the user's serving APs. Finally, a power control algorithm is proposed to improve the system throughput.

### 1.2. Related Works

In TDD CFMM systems, uplink training provides the downlink channel state information (CSI) by leveraging the channel reciprocity [8,9]. However, due to the limitation of the length of the coherent interval, pilots employed for channel estimation are non-orthogonal rather than orthogonal, which significantly affects the accurate estimation of the CSI and degrades system performance. This degradation in the system performance is named pilot contamination (PC). PC is an important factor that affects the ultimate limit of the system performance in TDD-based CFMM systems. Pilot allocation is considered an effective strategy to alleviate pilot contamination in massive MIMO systems. However, CFMM is different from massive MIMO [10] in terms of many aspects. Therefore, pilot allocation based on massive MIMO systems [13–15] cannot be directly applied to CFMM systems. Because of this, enormous efforts [16–19] have been made to analyze and design efficient pilot sharing in CFMM systems. In a random pilot allocation case, the pilot is assigned to users randomly. This leads to serious pilot interference when two users who share the same pilot are close. In [16], authors propose a scheme in which the minimum rate of all users is updated iteratively. This is called greedy pilot assignment. However, this solution only focuses on improving the rate of the worst terminals, rather than the overall performance. The simulation results in [16] show that there is still a significant gap between the random assignment and greedy assignment, and the orthogonal pilot assignment scheme. Recently, [17] proposed a pilot allocation scheme to maximize the minimum distance among users who share the same pilot; however, it is not easy to find centroid APs in CFMM. The authors in [18] improved greedy allocation according to location information to alleviate pilot contamination, in which users are assigned orthogonal pilots only when the number of pilots is large enough.

Furthermore, AP selection and power control play an important role in improving system performance [20–24]. For example, in terms of AP selection in a CFMM system, random AP selection has low complexity but cannot improve the throughput of the system. Reference [20] maximizes system throughput based on an AP selection algorithm according to the received signal power and path loss. In order to improve performance of users with poor communication environments, the transmit power of users or APs can be controlled to reduce interference and improve performance. A previous study [22] proposed a max–min power assignment algorithm to maximize the rate of the minimum user to improve the overall performance of the system. Reference [23] proposes an AP selection algorithm based on transmission power minimization and discusses the influence of path loss. Reference [24] controls the power of user pilots and uses the Taylor expansion to allocate the power of pilots, which improves system performance; however, the complexity of this proposal is too high.

### 1.3. Contributions

To alleviate the negative impact caused by pilot interference, we propose a joint resource allocation scheme including pilot allocation, AP selection, and power control. The main contributions of this paper are summarized as follows:

First, we introduce a low-complexity and effective pilot allocation scheme based on the order of users' distances to alleviate the negative impact of pilot interference. In this scheme, each user is sorted according to the distance to the center of the system, and then the pilot is assigned to users according to the sorting result.

Second, the AP selection algorithm is applied to reduce the system burden, where only those APs making larger contributions are selected as serving antennas. To further reduce the interference, we adjust the serving AP set to ensure that there are no common APs among users who share the same pilot.

Finally, the modified max–min power allocation scheme is jointly used to enhance system performance. The traditional max–min power assignment only focuses on improving the minimum rate, thus sacrificing the rate of other users, which degrades the system performance. By adjusting the objective function, the system throughput is added to the

objective function while considering maximizing the minimum rate. Then, the teaching and learning phases of the TLBO algorithm are used to maximize the total system throughput and the minimum rate.

The structure of this paper is arranged as follows. Section 2 depicts the system model of CFMM. In Section 3, we introduce resource allocation scheme for CFMM, including the UDOPA algorithm, AP selection, and modified max–min power allocation based on TLBO. In Section 4, we describe the simulation results and analysis. Section 5 presents the conclusions of this paper. The commonly used symbols are listed in Table 1.

**Table 1.** Notation used in this paper.

| | |
|---|---|
| $M$ | The number of APs |
| $\tau$ | The length of orthogonal pilots |
| $\boldsymbol{\varphi}_k$ | Pilot of the $k$-th user |
| $\mathbf{U}$ | Pilot matrix, each column is an orthogonal pilot sequence satisfying $\mathbf{U^H U=I}$ |
| $\mathbf{Ter}$ | Coordinate matrix of all users |
| A | Reference point which can be calculated by $K$-means method |
| $\mathbf{d}_k$ | The distance of the $k$-th user from point $A$ |
| $\overline{\mathbf{d}}_k$ | Rearrange $\mathbf{d}_k$ in ascending order, where $\overline{\mathbf{d}}_1$ indicates the closest distance between a user and the reference point |
| $K$ | The number of users |

## 2. System Model for CFMM

### 2.1. System Model

In this paper, we consider a TDD CFMM system with $M$ APs and $K$ users ($M > K$). APs and users are randomly located in a given field, as shown in Figure 1. For simplicity, we only consider the single antenna scenario. We assume that $g_{mk}$ represents the channel coefficient between the $k$-th user and the $m$-th AP. The model of $g_{mk}$ can be described as [16]:

$$g_{mk} = \beta_{mk}^{\frac{1}{2}} h_{mk} \tag{1}$$

where $h_{mk}$ represents the small-scale fading with i.i.d. $CN(0,1)$, and $\beta_{mk}$ represents the corresponding large-scale fading.

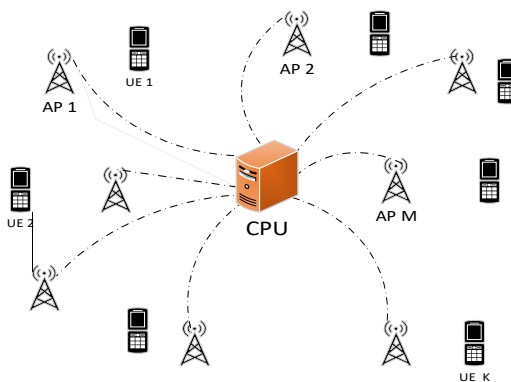

**Figure 1.** System model of the CFMM with $K$ users and $M$ APs.

### 2.2. Uplink Training

For CFMM systems, the first phase of the uplink process is uplink training. In this phase, each user simultaneously sends its pilot sequence $\sqrt{\tau}\boldsymbol{\varphi}_k \in \mathbf{C}^{\tau \times 1}$ to all APs for channel estimation, where $\|\boldsymbol{\varphi}_k\|^2 = 1$; $\tau$ is the length of the uplink training phase. However, the number of orthogonal pilots depends on the length of the uplink training phase which is much smaller than $K$. Therefore, pilot reuse is necessary for practical applications. Pilot interference occurs when pilot reuse is allowed in the system, which significantly reduces

the performance of system. To mitigate the pilot interference, we describe and analyze the uplink transmission model to establish the mathematical framework for the pilot allocation scheme. In this phase, the $m$-th AP receives:

$$y_{p,m} = \sqrt{\tau\rho_p}\sum_{k=1}^{K} g_{mk}\boldsymbol{\varphi}_k + w_{p,m} \tag{2}$$

where $\rho_p$ is the normalized signal to noise ratio (SNR) and $w_{p,m}$ is an AWGN with i.i.d. $CN(0,1)$. According to $y_{p,m}$ in Equation (2), we estimate $g_{mk}$ as [16]:

$$\begin{aligned}\hat{g}_{mk} &= \frac{E\{\hat{y}_{p,mk}g_{mk}\}}{E\{|\hat{y}_{p,mk}|^2\}}\hat{y}_{p,mk}\\ &= c_{mk}\hat{y}_{p,mk}\end{aligned} \tag{3}$$

where:

$$c_{mk} = \frac{\sqrt{\tau\rho_p}\beta_{mk}}{\tau\rho_p\sum_{k=1}^{K}\beta_{mk}|]\boldsymbol{\varphi}_k^H\boldsymbol{\varphi}_k| + 1} \tag{4}$$

with:

$$\hat{y}_{p,mk} = \boldsymbol{\varphi}_k^H y_{p,m}$$

### 2.3. Uplink Data Transmission

In the second phase of the uplink process, i.e., uplink data transmission, the $m$-th AP receives [16]:

$$y_{u,m} = \sqrt{\rho_u}\sum_{k=1}^{K} g_{mk}\sqrt{\eta_k}q_k + w_{u,m} \tag{5}$$

where $q_k$ represents the transmit signal from the $k$-th user where $E\{|q_k|_2\} = 1$ and $\eta_k$ represents the data power coefficient with $\eta_k \in [0,1]$; $w_{u,m} \sim CN(0,1)$. In addition, $\rho_u$ is the normalized uplink SNR.

In order to decode $q_k$ from user $k$, the $m$-th AP sends $\hat{g}_{mk}^* y_{u,m}$ to the CPU over the fronthaul network using the maximum ratio (MR) combiner. Then, CPU receives:

$$r_{u,k} = \sum_{m\in\Lambda_k}\hat{g}_{mk}^* y_{u,m} = \sum_{k=1}^{K}\sum_{m\in\Lambda_k\cap\Lambda_{k'}}\sqrt{\rho_u\eta_k}\hat{g}_{mk}^* g_{mk}q_k + \sum_{m\in\Lambda_k}\hat{g}_{mk}^* w_{u,m} \tag{6}$$

where $\Lambda_k$ is the serving AP set of the $k$-th user, and $\Lambda_{k'}$ is the serving AP set of the $k'$-th user. Then, the achievable throughput for each user is calculated as [16]:

$$R_{u,k} = \log_2\left(1 + \frac{\rho_u\eta_k\left(\sum\limits_{m=1}^{M}\gamma_{mk}\right)^2}{\rho_u\sum\limits_{k'\neq k}^{K}\eta_{k'}\left(\sum\limits_{m=1}^{M}\gamma_{mk}\frac{\beta_{mk'}}{\beta_{mk}}\right)^2|\boldsymbol{\varphi}_k^H\boldsymbol{\varphi}_{k'}|^2 + \rho_u\sum\limits_{k'=1}^{K}\eta_{k'}\sum\limits_{m=1}^{M}\gamma_{mk}\beta_{mk'} + \sum\limits_{m=1}^{M}\gamma_{mk}}\right) \tag{7}$$

According to Equation (7), the achievable throughput with AP selection can be calculated as:

$$R_{u,k} = \log_2\left(1 + \frac{\rho_u\eta_k\left(\sum\limits_{m\in M_k}\gamma_{mk}\right)^2}{\rho_u\sum\limits_{k'\neq k}^{K}\eta_{k'}\left(\sum\limits_{m\in\Lambda_k\cap\Lambda_{k'}}\gamma_{mk}\frac{\beta_{mk'}}{\beta_{mk}}\right)^2|\boldsymbol{\varphi}_k^H\boldsymbol{\varphi}_{k'}|^2 + \rho_u\sum\limits_{k'=1}^{K}\eta_{k'}\sum\limits_{m\in\Lambda_k\cap\Lambda_{k'}}\gamma_{mk}\beta_{mk'} + \sum\limits_{m\in\Lambda_k}\gamma_{mk}}\right) \tag{8}$$

Notice that, if the set of serving APs includes all of the APs, Equation (8) will be simplified to Equation (7). It can be seen from Equation (8) that only a portion of the APs

participate in providing the service for users, which greatly reduces the computational complexity of the system.

## 3. Resource Allocation for CFMM

In this section, we provide details of the proposed radio resource allocation scheme, which includes the user-distance-ordering-based pilot assignment (UDOPA) strategy, AP selection, and a modified max–min power allocation.

### 3.1. The UDOPA

The main idea of UDOPA can be described as follows: First, find the center of the network as the reference point in the UDOPA algorithm which can be obtained using the *K*-means method. Second, allocate the pilot according to the distance between the user and the reference point. This step ensures that two users who share the same pilot are not too close to each other.

Specifically, the CPU first calculates the distance from each user to the reference point according to the matrix **Ter** and obtains a distance vector $\mathbf{d}_k$. The vector $\mathbf{d}_k$ is listed in ascending order, and we can obtain a new vector denoted $\overline{\mathbf{d}}_k$. The CPU performs the pilot assignment task repeatedly until each user is assigned a pilot. The user with a vector $\overline{\mathbf{d}}_{n-\tau+k}$ will be allowed to use the *k*-th column of the pilot sequence in matrix U. Here, *n* is a natural number.

In Figure 2, an example is shown for the pilot allocation when *K* = 8 and $\tau$ = 4, where the graphs with the same shape indicate users assigned to the same pilot, and the numbers next to the graphs represent the order of the distance from the user to the center point. For example, two solid circle users in the figure are allowed to share the pilot.

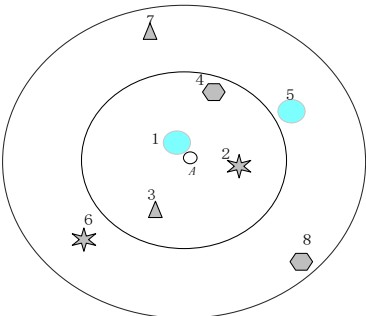

**Figure 2.** An example of the UDOPA with 4 orthogonal pilots.

### 3.2. AP Selection Based on UDOPA

To reduce the complexity, we attempt to select a portion of the APs as the serving APs. Similar to [19,20], the AP selection is based on large-scale fading coefficients. We select $|\Lambda_k|$ APs as the serving APs for user *k* if:

$$\sum_{m \in \Lambda_k} \frac{\overline{\beta}_{mk}}{\sum\limits_{m=1}^{M} \beta_{mk}} \geq \theta \tag{9}$$

Here, we sort $\{\beta_{1k}, \ldots, \beta_{mk}\}$ in descending order and obtain a new variable $\{\overline{\beta}_{1k}, \ldots, \overline{\beta}_{Mk}\}$. $\theta$ is a constant which indicates that at least $\theta$ of the total received power of the desired signal is contributed to the *k*-th user with only $|\Lambda_k|$ APs. Then, we obtain a matrix $\mathbf{\Psi}$. In the matrix $\mathbf{\Psi}$, each row represents the subset of APs selected by each user. According to the matrix $\mathbf{\Psi}$, we can obtain a binary matrix $\mathbf{\Psi}$. Element 1 of the matrix $\mathbf{\Psi}$ means that the user selects the corresponding AP as its serving AP, and the element 0 means that the user will not select the AP as its serving AP. In contrast from the existing AP selection algorithms, users using

the same pilot will not allowed to share the same serving AP in our AP selection scheme. That is, if two users who share the same pilot have public serving APs, one of them will delete these public APs from its serving AP set.

The detailed procedure is described in Algorithm 1.

---

**Algorithm 1:** Proposed resource allocation with UDOPA and AP selection

---

Input: location of APs, location of users, $M$, $K$, $\tau$.

1. **while** there is unassigned user do;
2. Calculate the distance $\mathbf{d}_k$ between each user $k$ and point $A$, sort $\mathbf{d}_k$ in ascending order and denote them $\bar{\mathbf{d}}_k$;
3. Assign pilots according to $\bar{\mathbf{d}}_k$;
4. The user with a vector $\bar{\mathbf{d}}_{n-\tau+k}$ will be allowed to use the $k$-th column of the pilot sequence in matrix $\mathbf{U}$. Here, $n$ is a natural number;
5. **end while**;
6. According to (14), select the $\Lambda_k$ APs for the $k$-th user and create a binary matrix $\breve{\mathbf{\Psi}}$ which indicates whether an AP is selected by a user;
7. **while** $\Lambda_k \cap \Lambda_{k'} \neq \varnothing$ (assume that user $k$ and user $k'$ use the same pilot);
8. Remove a public AP from the set $\Lambda_k$ if user $k$ has more serving APs than user $k'$; otherwise, remove the public AP from the set $\Lambda_{k'}$. Then update the matrix $\breve{\mathbf{\Psi}}$;
9. **end while**.

Output: the matching of AP–user–pilot.

---

Figure 3 describes the process of UDOPA and AP selection. In Figure 3a, UEs having the same color are allowed to share the same pilot in the system after the UDOPA algorithm. The APs in the circle are serving APs of the UE in this circle. From Figure 3a, the UEs having a red color have two public Aps, which will cause serious pilot interference and reduce the system performance. By using the modified AP selection based on Algorithm 1 as shown in Figure 3b, we prevent the red UEs from sharing the same AP.

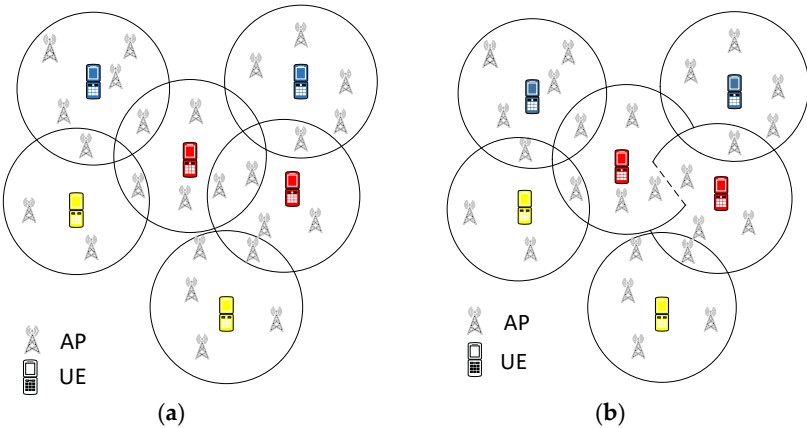

(**a**)    (**b**)

**Figure 3.** UDOPA combined with AP selection: (**a**) AP selection based on (9); (**b**) the modified AP selection based on (8) and (9) in Algorithm 1.

### 3.3. Modified Max–Min Power Allocation for CFMM Based on TLBO

3.3.1. Modified Max–Min Objective Function for CFMM

Power control is a popular research topic in CFMM systems. In order to provide uniformly good service to all users, we must ensure the minimum rate for all users. In this subsection, we discuss the uplink power control for CFMM systems, i.e., provide uniform services to users while ensuring the maximum overall system rate.

In general, the max–min power assignment can be expressed as:

$$\max_{\{\eta_k\}} \min_{k=1,\ldots,K} R_{u,k}$$
$$subject\ to\ 0 \leq \eta_k \leq 1, k = 1, \ldots, K \tag{10}$$

where $R_{u,k}$ is given by Equation (8). Equation (10) shows that the max–min power assignment problem focuses on maximizing the minimum rate, so it will decrease other users' performance in the system, thus reducing the total system throughput. To address the shortcomings of the max–min power assignment, Equation (10) is reformulated as:

$$\max_{\{\eta_k\}} \sum_{k=1}^{K} R_{u,k} \times \min_{k=1,\ldots,K} R_{u,k}$$
$$subject\ to\ 0 \leq \eta_k \leq 1, k = 1, \ldots, K \tag{11}$$

In Equation (11), we add the total system throughput into the objective function so that the total system throughput can be maximized while ensuring the minimum rate. To solve Equation (10), a teaching–learning-based optimization (TLBO) is proposed as follows.

### 3.3.2. TLBO Scheme

The TLBO scheme is conducted based on an instructional design in which students need to be taught by the teacher to improve their learning performance. They also learn from each other to facilitate the absorption of knowledge. Here, the teacher is the one who performs best among individuals and the course chosen by each student is a decision variable [25].

For the optimization objective: $z = \max\{f(\mathbf{x})|\mathbf{x} \in S\}$, the search range $S = \{\mathbf{x}|x_i \in (x_i^L, x_i^U), i = 1, 2, \ldots, d\}$, with $\mathbf{x} = (x_1, x_2, \ldots, x_d)$, the subscript $d$ indicates the dimension of the search range, $x_i^L$ and $x_i^U$ denote the maximum and minimum values of each $x_i$ desirable value in the dimension, respectively. $f(\mathbf{x})$ is the objective function. We assume that $\mathbf{x}^j = \left(x_1^j, x_2^j, \ldots, x_d^j\right)$ is a position in the search range, $x_i^j(i = 1, 2, \ldots, d)$ is the decision factor in $\mathbf{x}^j$, and $N$ is the number of points in the search range. Finally, a class can be represented by Equation (12):

$$\begin{bmatrix} \mathbf{x}^1 & f(\mathbf{x}^1) \\ \mathbf{x}^2 & f(\mathbf{x}^2) \\ \vdots & \vdots \\ \mathbf{x}^N & f(\mathbf{x}^N) \end{bmatrix} = \begin{bmatrix} x_1^1 & x_2^1 & \cdots & x_d^1 & f(\mathbf{x}^1) \\ x_1^2 & x_2^2 & \cdots & x_d^2 & f(\mathbf{x}^2) \\ \vdots & \vdots & & \vdots & \vdots \\ x_1^N & x_2^N & \cdots & x_d^N & f(\mathbf{x}^N) \end{bmatrix} \tag{12}$$

1.  Class—the set of all students;

2.  Learner—the $j$-th learner in the class is recorded as $\mathbf{x}^j = \left(x_1^j, x_2^j, \ldots, x_d^j\right)$;

3.  Teacher—the teacher $\mathbf{x}_{teacher}$ is the top one in the class.

A.  Teacher phase in TLBO Algorithm

During this phase, learners learn from the teacher who tries to increase the mean value of the whole class.

For the objective function $f(\mathbf{x})$, $\mathbf{x}$ is a $d$-dimensional decision factor, then the $j$-th learner can be denoted as $\mathbf{x}^j = \left(x_1^j, x_2^j, \ldots, x_d^j\right)$. For a class with $N$ students, each student will update his or her position by:

$$\mathbf{x}_{new}^j = \mathbf{x}_{old}^j + r_j * (\mathbf{x}_{teacher} - T_F \mathbf{x}_{mean}) \tag{13}$$

$$\mathbf{x}_{mean} = \frac{1}{N} \left[ \sum_{i=1}^{N} x_1^i, \sum_{i=1}^{N} x_2^i, \ldots, \sum_{i=1}^{N} x_d^i \right] \tag{14}$$

where $\mathbf{x}_{new}^{j}$ and $\mathbf{x}_{old}^{j}$ represent the updated position and the original position of the *j*-th learner, respectively, $r_j \in [0, 1]$ is a random number, $\mathbf{x}_{teacher}$ denotes the teacher, $T_F$ is the teaching factor, and:

$$T_F = round[1 + rand(0,1)] \tag{15}$$

For the *j*-th learner, $\mathbf{x}_{new}^{j}$ is accepted if $f\left(\mathbf{x}_{new}^{j}\right)$ is better than $f\left(\mathbf{x}_{old}^{j}\right)$; otherwise, $\mathbf{x}_{new}^{j}$ is rejected.

B.　Learner phase

During this phase, each learner $\mathbf{x}^j (j = 1, 2, \ldots, N)$ randomly selects a learner $\mathbf{x}^i$ $(i = 1, 2, \ldots, N, i \neq j)$ in the class. The learner $\mathbf{x}^j$ updates its position by analyzing the difference with learner $\mathbf{x}^i$ in the following way:

$$\mathbf{x}_{new}^{j} = \begin{cases} \mathbf{x}_{old}^{j} + r_j * \left(\mathbf{x}^j - \mathbf{x}^i\right), f\left(\mathbf{x}^j\right) < f\left(\mathbf{x}^i\right) \\ \mathbf{x}_{old}^{j} + r_j * \left(\mathbf{x}^i - \mathbf{x}^j\right), else \end{cases} \tag{16}$$

where $r_j$ is the learning factor of the *j*-th learner.

Similar to the teacher phase, $\mathbf{x}_{new}^{j}$ is accepted if $f\left(\mathbf{x}_{new}^{j}\right)$ is better than $f\left(\mathbf{x}_{old}^{j}\right)$; otherwise, $\mathbf{x}_{new}^{j}$ is rejected.

### 3.3.3. Modified Max–Min Power Control Based on TLBO

Based on the previous description, we set the search range $S = \left\{\mathbf{x} | x_i \in \left(x_i^L, x_i^U\right), i = 1, 2, \ldots, d\right\}$ as $S = \left\{\mathbf{x} | \eta_i \in \left(x_i^L, x_i^U\right), i = 1, 2, \ldots, K\right\}$. Each decision factor in the learner corresponds to the power allocation factor for each user in the uplink, and the number of decision factors is kept consistent with the number of users *K* in the CFMM system, i.e., $\mathbf{x} = (\eta_1, \eta_2, \ldots, \eta_k)$. Set $x_i^L = 0, x_i^U = 1$ $(i = 1, 2, \ldots, K)$. The final objective function is set as:

$$f(\mathbf{x}) = \sum_{k=1}^{K} R_{u,k} \times \min_{k=1,\ldots,K} R_{u,k} \tag{17}$$

The max–min power allocation maximizes only the minimum rate, and therefore reduces other users' rates. To solve this problem, we propose a modified max–min objective function in Equation (11). Based on the introduction of the TLBO in Section 3.3.2, the modified max–min power allocation based on TLBO for CFMM systems can be summarized in Algorithm 2.

---

**Algorithm 2:** The proposed power control algorithm.

---

Input: *N* (number of learners) and *K* (dimension, i.e., number of users).

1.　Initialization: randomly assign power control coefficients to learners with $\mathbf{x} = (\eta_1, \eta_2, \ldots, \eta_k)$.
2.　**For** the teacher phase,
3.　Find $\mathbf{x}_{teacher} = \text{argmax} f(\mathbf{x})$ and calculate the mean of all learners by (14).
4.　Calculate $TF = round(1 + rand(0,1))$ and $\mathbf{x}_{new}^{j}$ by (12)–(14).
5.　**If** $f\left(\mathbf{x}_{new}^{j}\right) > f\left(\mathbf{x}_{old}^{j}\right)$, update the current solution to $\mathbf{x}_{new}^{j}$; otherwise, reject $\mathbf{x}_{new}^{j}$.
6.　**For** the learner phase,
7.　**For** each learner, randomly select a learner and update the position according to (16).
8.　If $f\left(\mathbf{x}_{new}^{j}\right) > f\left(\mathbf{x}_{old}^{j}\right)$, update the current solution to $\mathbf{x}_{new}^{j}$; otherwise, reject $\mathbf{x}_{new}^{j}$.
9.　Return the optimal power control factor $\mathbf{x}^*$
10.　**Stop if** the stop condition is satisfied; otherwise, go to step 2.

Output the optimal power control factor $\mathbf{x}^*$.

---

## 4. Numerical Results

The simulation scenario is a square area of $1 \times 1$ km$^2$. The noise power is given by $P_n = Bk_bT_0W$ where $k_b$ is the Boltzmann constant, and $T_0 = 290$. The simulation parameters of the system are set as shown in Table 2.

**Table 2.** System parameters.

| Parameters | Value |
|---|---|
| Carrier frequency: $f$ | 1.9 GHz |
| Bandwidth: $B$ | 20 MHz |
| User and AP antenna height : $h_m$, $h_b$ | 1.65 m, 15 m |
| Pilot sequence length : $\tau$ | 20 |
| $\rho_d$ | 200 mw |
| $\rho_p$ | 100 mw |
| $\rho_u$ | 100 mw |
| $\sigma_{sh}$ | 8 dB |
| Noise figure | 9 dB |
| $d_1$ | 50 m |
| $d_0$ | 10 m |
| $D$ | 1000 m |

$\beta_{mk}$ in (11) is expressed as:

$$\beta_{mk} = PL_{mk}10^{\frac{\sigma_{sh}Z_{mk}}{10}} \tag{18}$$

where $PL_{mk}$ is the path loss, and $10^{\frac{\sigma_{sh}Z_{mk}}{10}}$ means shadow fading with a standard deviation $\sigma_{sh}$ and $Z_{mk} \sim N(0,1)$. Similar to [8], $PL_{mk}$ is calculated by a three-slope model. Here, the throughput is calculated as $S_{u,k}^{cf} = B\frac{1-\frac{\tau}{\tau_c}}{2}R_{u,k}^{cf}$ where $\tau_c = 200$ samples.

*Results and Discussions*

Figure 4 describes the cumulative distribution function (CDF) curve of the uplink throughput when the number of APs is 100 and 300, respectively. As observed, performance of UDOPA is superior to that of random pilot allocation (RPA), greedy pilot allocation (GPA), and local-based GPA (LBGPA), and is close to the orthogonal pilot assignment with no pilot contamination (NoPC). At the same time, the UDOPA achieves 93% of the NoPC in terms of 95%-likely per-user throughput. The reason for this is that we effectively avoid both severe pilot contamination and the impact of using the same APs on system performance. Figure 4 also shows that the system throughput increases with the increase in the number of APs.

Figure 5 shows uplink throughput vs. the number of pilots, $\tau$. As observed, the average throughput decreases with the increase in $\tau$. This is because, as the number of orthogonal pilots increases, the pilot contamination decreases, yet the channel estimation overhead increases, and the reduction in pilot contamination is not sufficient to compensate for the channel estimation overhead. As observed, compared to the other three schemes (RPA, LBGPA, and GPA), the proposed UDOPA has the best average uplink throughput performance. Although the performance of UDOPA is worse than that of NoPC, it is almost impossible for NoPC to exist in a practical scheme.

Figure 6 shows the average throughput vs. $\theta$. It shows that, as the value of $\theta$ increases, the average throughput first rises to a peak, and then decreases. Here, $\theta = 0.98$ corresponds to the case in which 13 APs are chosen to serve each user instead of 100, which greatly reduces the complexity of the system.

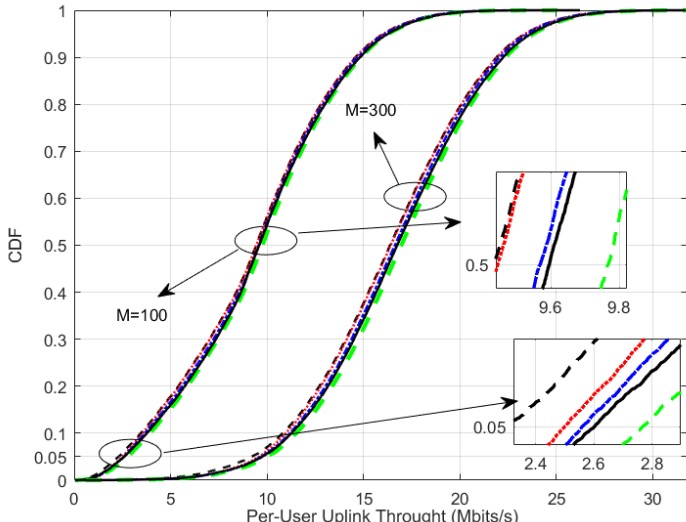

**Figure 4.** The CDF of the uplink throughput (the curves from left to right: RPA, GPA, LBGPA, UDOPA, NoPC).

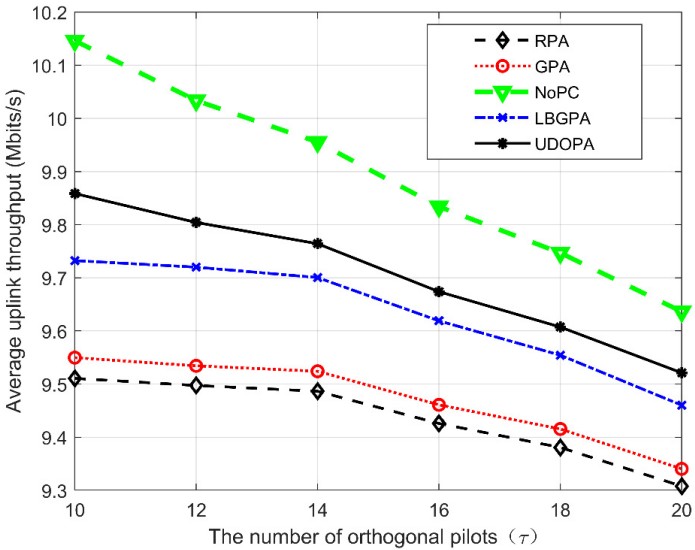

**Figure 5.** Throughput of the system vs. $\tau$.

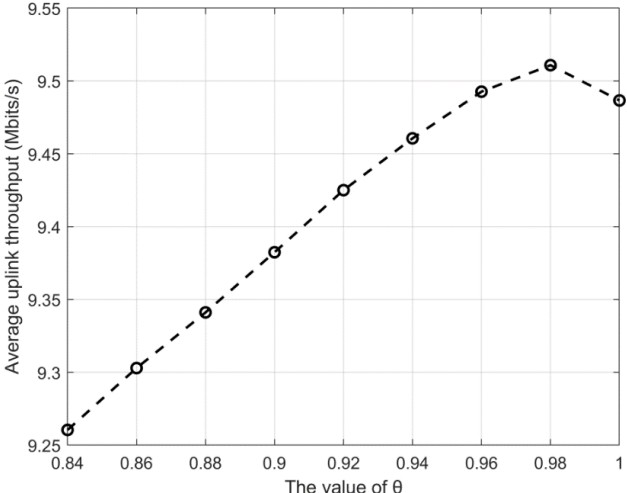

**Figure 6.** The average throughput of the system vs. $\theta$.

Figure 7 describes the CDF of throughput with different power control schemes where TLBO power control is the modified max–min power control based on TLBO. From this figure, we can see that there are many low-rate users in the scenario with no power control. TLBO power control and max–min power assignment ensure a minimum rate for users, and thus both achieve a uniform quality of service for users. In terms of 95% user throughput, TLBO-based control performs slightly better than max–min power assignment.

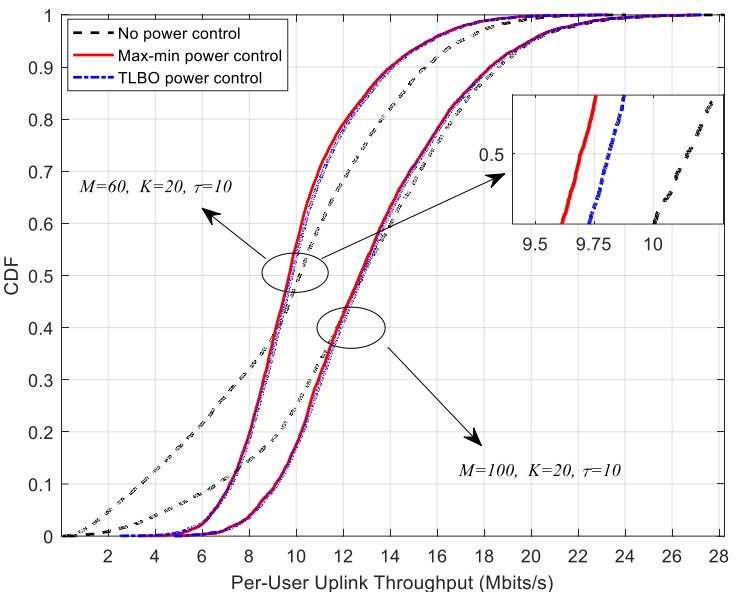

**Figure 7.** CDF of throughput per user with different power controls.

Figure 8 shows that the average throughput of the proposed TLBO power control is superior to that of the NoPC case and the case with max–min power assignment. Compared with the max–min power assignment, the average throughput of our algorithm increases by 0.1075 Mbits/s and the total system throughput increases by 2.15 Mbits/s. It also shows that the TLBO power control further improves the system performance while ensuring the minimum rate.

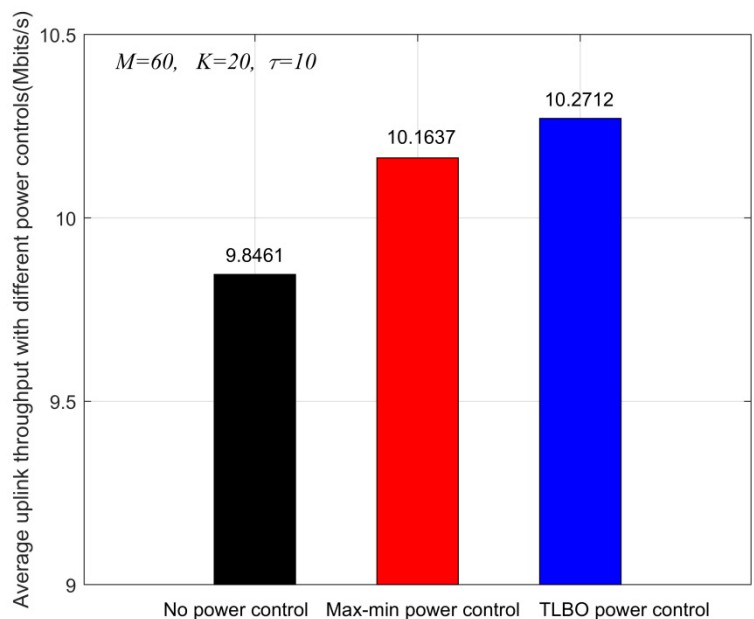

**Figure 8.** Average uplink throughput with different power controls.

## 5. Conclusions

We propose a joint optimization scheme including pilot allocation, AP selection, and power control to improve throughput of CFMM systems. First, an effective pilot allocation strategy is proposed, where an orthogonal pilot sequence is allocated to users having a small user–user distance. Pilot sharing is only allowed when the distance between two users is large enough. Then, AP selection is performed based on large-scale fading coefficients, where important APs are selected to serve the users. In addition, modified max–min power control with the TLBO algorithm is proposed to further improve the system performance. The simulation results show that the joint optimization scheme proposed in this paper significantly improves the throughput and the proposed UDOPA with AP selection effectively mitigates pilot interference. It also shows that effective power allocation, such as via the TLBO algorithm, will significantly improve the throughput. In this paper, we only consider the single antenna scenario. In future work, multiple antennas can be considered in the CFMM system to enhance the performance. Furthermore, the system discussed in this paper works in TDD mode, and the system performance in FDD mode can be explored in future work.

**Author Contributions:** Conceptualization, F.X. and X.L.; methodology, software, X.L. and Y.W.; validation, J.F. and Y.W. All authors have read and agreed to the published version of the manuscript.

**Funding:** This work is supported by Natural Science Foundations of Zhejiang Province, Grant No. LY20F010009 and LY22F010012.

**Conflicts of Interest:** The authors declare no conflict of interest.

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
