# Peer review of "Resource Allocation for TDD Cell-Free Massive MIMO Systems"

_electronics, doi:10.3390/electronics11121914_

Round 1
Reviewer 1 Report
In this paper, the authors present a joint optimization scheme including pilot allocation, AP selection and power control for Cell-Free Massive MIMO systems. It is shown in the paper, by numerical results, that this method improves the system throughput. First, the authors describe an effective pilot allocation strategy, where orthogonal pilot sequence is allocated to users with small user-user distance. Then, AP selection is done based on large-scale fading coefficients. Also, to improve the system performance, the authors present a modified max-min power control with TLBO algorithm.
In order to improve the paper, the authors must consider the following:
- In the section 2 to present clearly what is their work and what is taken from literature. There are a lot of equations without citation
- Review the English, there are a lot of paragraphs with minor error (line 114, 116 etc.)
- - The TLBO algorithm is presented in section 3.2.2 without any citation
- - In table 2 is GHz and MHz no GHZ and MHZ
- - All the abbreviations in the text must be explained (CDF, etc.)
- - In figure 6 on the Ox axis is θ
- - The reference 21 don’t exist in the text
Reviewer 2 Report
The paper proposes and investigates a joint resource allocation problem in time-division duplex (TDD) based cell-free massive MIMO (CFMM) systems which could improve spectrum efficiency and throughput. Among others, a pilot assignment scheme based on the distance between the users and the center is proposed together with a modified max-min power control scheme with teaching-learning-based optimization.
The paper provides sufficient background information and the relevant literature as well as a generally satisfactory set of simulation results.
The full term of UDOPA should be given in the text.
Lines 60-65, that present the paper’s topic, should be enriched. It could be possibly replaced by the abstract which provides a more complete description of the paper’s subject.
Figure 5 (lines 240-244) is not adequately analyzed; it should include more details, e.g. a comparison between the involved schemes.
In figure 6 (horizontal axis) “ζ” should be corrected to “θ”.
The “Conclusions” section is rather short and not adequately informative. For example, it could include an evaluative overview of the obtained results as well as a reference to possible future work.
All in all, the paper is publishable subject to the comments made above and a moderate editing regarding the use of English. I would consider the revision as "moderate", however, since such a category does not exist, I characterize it as "major".
Reviewer 3 Report
The paper in general is written well. However, it is recommended that the authors’ should perform the following
1. Improve the writing discrepancies, e.g. Line 59 sentence structure, etc.
2. The paper lacks in the motivation aspect, perhaps more effort should be placed to justify the rational of work.
3. The contributions of the paper are not clear enough. Please enumerate them in the introduction section.
4. Paper has notational issues and inconsistency. I will suggest to use bold small letters for vectors, italic small letters for scalars, and bold capital letters for matrices. Also, define the notations used in the paper either as a footnote, or at the end of introduction section. Use the proper notation for the expectation operator and also define it.
5. Equation (4) needs to be rewritten properly.
6. Elaborate clearly on how you arrived at (8) from (7).
7. Algorithm I and II shall start by mentioning what are input and what are output.
8. Results of CDF: the y-axis should have the range 0 to 1 rather that 0 to 0.7.
9. Caption of the Figures needs more elaborations.
10. In the body of the paper, authors have made links to complexity of Algorithms, however, in the results part this angle is missing.
Round 2
Reviewer 2 Report
I am satsfied with the authors' response, so my recommendation is the paper to be accepted in basically its present form, perhaps after a typical (minor) editing regarding the use of English.
Reviewer 3 Report
Authors have addressed my comments well. The paper can be accepted for publication.